# The Drug Transporter P-Glycoprotein and Its Impact on Ceramide Metabolism—An Unconventional Ally in Cancer Treatment

**DOI:** 10.3390/ijms25189825

**Published:** 2024-09-11

**Authors:** Johnson Ung, Miki Kassai, Su-Fern Tan, Thomas P. Loughran, David J. Feith, Myles C. Cabot

**Affiliations:** 1Department of Microbiology, Immunology, and Cancer Biology, University of Virginia School of Medicine, Charlottesville, VA 22908, USA; ju7a@virginia.edu; 2Department of Biochemistry and Molecular Biology, Brody School of Medicine, East Carolina University, The East Carolina Diabetes and Obesity Institute, Greenville, NC 27834, USA; kassaim18@ecu.edu; 3University of Virginia Cancer Center, University of Virginia School of Medicine, Charlottesville, VA 22908, USA; st4ub@virginia.edu (S.-F.T.); djf2g@virginia.edu (D.J.F.); 4Department of Medicine, Hematology/Oncology, University of Virginia School of Medicine, Charlottesville, VA 22908, USA

**Keywords:** ceramide, sphingolipids, cancer drug resistance, P-glycoprotein

## Abstract

The tumor-suppressor sphingolipid ceramide is recognized as a key participant in the cytotoxic mechanism of action of many types of chemotherapy drugs, including anthracyclines, Vinca alkaloids, the podophyllotoxin etoposide, taxanes, and the platinum drug oxaliplatin. These drugs can activate de novo synthesis of ceramide or stimulate the production of ceramide via sphingomyelinases to limit cancer cell survival. On the contrary, dysfunctional sphingolipid metabolism, a prominent factor in cancer survival and therapy resistance, blunts the anticancer properties of ceramide-orchestrated cell death pathways, especially apoptosis. Although P-glycoprotein (P-gp) is famous for its role in chemotherapy resistance, herein, we propose alternate interpretations and discuss the capacity of this multidrug transporter as a “ceramide neutralizer”, an unwelcome event, highlighting yet another facet of P-gp’s versatility in drug resistance. We introduce sphingolipid metabolism and its dysfunctional regulation in cancer, present a summary of factors that contribute to chemotherapy resistance, explain how P-gp “neutralizes” ceramide by hastening its glycosylation, and consider therapeutic applications of the P-gp-ceramide connection in the treatment of cancer.

## 1. Sphingolipid Metabolism at a Glance

### 1.1. The De Novo Pathway of Ceramide Formation

The study of lipid metabolism was likely not a favorite in school, and sphingolipids are no exception. They are numerous, of complex structure and nomenclature, and of intricate metabolism, especially regarding the attached sugar moieties of glycosphingolipids. Most unfamiliar with the field will know the star of this review, ceramide, as a component of hand lotion and/or as a factor in the skin that moisturizes and forms a protective barrier. But there is much more to the story.

Ceramide and higher sphingolipids play a big role in cancer biology. These molecules comprise a very large family of lipids found in animals, plants, fungi, some prokaryotic organisms, and viruses and do not contain a glycerol backbone common to glycerolipids (mono-, di-, and tri-acylglycerols, phospholipids). Instead, sphingolipids contain a sphingoid backbone, sphinganine (Figure 1), that can be acylated at the amide linkage to form dihydroceramide, which can be desaturated to generate ceramide. Ceramide is the building block of sphingolipids or, if they contain sugars, glycosphingolipids. These lipids serve structural roles in bio-membranes and have wide-ranging effects on signaling and regulation of cell function. Chief among these actions is the potentiation of signal transduction events that promote cell death, which we call the “ceramide effect”.

The steps of de novo ceramide biosynthesis are illustrated in Figure 1. Ceramide synthesis starts with serine palmitoyltransferase. This enzyme uses serine and palmitoyl-CoA to generate the ‘sphingoid’ base, sphinganine. Sphinganine is then acylated by dihydroceramide synthase, a sextet of enzymes (CerS1-6) that produce a wide array of chemically diverse dihydroceramides via their preferences for specific chain length acyl-CoA’s [1,2,3]. This chain-length specificity produces molecular species with carbon chain lengths ranging from C14 to C32, with varying degrees of saturation. The penultimate step in ceramide production is catalyzed by dihydroceramide desaturase via the insertion of a 4,5-*trans* double bond. Ceramide can also be produced by the action of sphingomyelinases. These enzymes, which are classified according to their optimum pH and subcellular locations, cleave sphingomyelin at the phosphodiester bond that is proximal to ceramide, producing ceramide and choline phosphate. Ceramides also accumulate via ceramidase inhibition, of which there are five ceramidase isoforms that vary by subcellular localization and pH-dependent functionality [4,5], and glucocerebrosidase activity, the back-reaction to glucosylceramide synthase [6,7] (Figure 1).

### 1.2. Ceramide Metabolism—Intracellular Fate

Once generated, what is the metabolic fate of ceramide? By and large, tumor cells can be protected from ceramide-induced apoptosis by major metabolic pathways that divert ceramide to a wide variety of sphingolipids such as sphingomyelin (SM) and glucosylceramide (GlcCer) (Figure 1). GlcCer is the precursor of higher cerebrosides, globosides, and gangliosides, whereas galactosylceramides serve to produce sulfatides. For information on inhibitors and activators of sphingolipid metabolism, see an earlier review by Bleicher and Cabot [8].

Of prime interest in this review is ceramide glycosylation, a step that is assisted by the drug resistance pump, P-glycoprotein (P-gp), although the enzyme responsible for the production of GlcCer is glucosylceramide synthase (GCS). Below, we use the terms ceramide glycosylation and GlcCer production interchangeably to reference the same pathway. P-gp acts as a conduit for GlcCer and, in this manner, assists in its removal from GCS, a step that facilitates GlcCer production. Ceramide glycosylation is the first step in the production of neutral glycosphingolipids. However, in cancer cells, specifically multidrug-resistant cancer, glycosylation is chiefly utilized to “neutralize” ceramide, effectively mitigating ceramide-driven apoptosis [8,9,10,11,12]. Whereas ceramide is a powerful tumor suppressor, the glycosylated product GlcCer is ineffectual in this realm. In summary, upregulated ceramide glycosylation is an avenue through which cancer cells skirt the deleterious effects of high intracellular ceramide concentrations [13,14,15,16,17]. 

Also relevant in bypassing ceramide-elicited apoptosis is ceramide hydrolysis, specifically by acid ceramidase (AC), another sentinel enzyme regulator of cancer cell growth [4,18,19,20,21,22]. Similar to GCS, AC thwarts the tumor-toxic properties of ceramide via hydrolysis, producing a fatty acid and sphingosine, the latter a substrate for sphingosine kinase (SPHK), the enzyme catalyzing the production of sphingosine-1-phosphate (S1-P), a cancer cell mitogen [23,24,25]. In these instances, we can envision the delicate balance that must be maintained between the levels of ceramide, controlled by GCS and AC, and the levels of S1-P, controlled by AC and SPHK. Also working to decrease endogenous ceramide levels are the sphingomyelin synthases and ceramide kinase that catalyze sphingomyelin synthesis and ceramide phosphorylation, respectively [26,27,28,29]. Ceramide kinase has gained recognition in the cancer biology field [30]. This enzyme generates ceramide 1-phosphate, and its inhibition drives ferroptosis and enhances sensitivity to cisplatin in lung cancer models [31].

### 1.3. Bioactive Sphingolipids

Ceramide and S1-P, termed bioactive sphingolipids, play an imposing role in cancer cell fate, as they are implicated in a myriad of processes, including apoptosis, ferroptosis, autophagy, senescence, growth inhibition, and mitogenicity [32,33,34]. The intracellular accumulation of long-chain ceramides (C16 or C18) elicits apoptosis; however, this can be circumvented by the action of ceramide-clearance enzymes that glycosylate, phosphorylate, and hydrolyze ceramide. This metabolism marks a shift toward “drug resistance”, a biology that we demonstrated when selecting cancer cells for resistance to various chemotherapy drugs [35,36]. In contrast to ceramide, S1-P is a pro-survival sphingolipid that conducts mitogenic responses. Many see the balance between ceramide and S1-P as critical in the control of cancer cell fate, and thus, factors that impact levels of the two can be employed toward therapeutic benefit [37,38,39].

As nature would have it, the actions of the various ceramides are molecular species-specific and influence diverse biology ranging from cancer to diabetes to coronary artery disease to cystic fibrosis [3,40,41,42,43,44,45,46,47]. For example, in aging, where the loss of skeletal muscle mass is widespread, CerS1 and CerS5 were identified as potential regulators of fiber size and strength [48]. In cancer, work by Siddique et al. [49] illustrated that ceramide molecular species nuances focusing on the 4,5-*trans* double bond were required for cancer cell sensitivity to etoposide. Rudd and Devaraj [50] showed that the apoptotic impact of several ceramide species was dependent on acyl-chain saturation. For example, fully saturated ceramides, C16:0 and C18:0, significantly reduced viability, in contrast with C18:1 and C24:1, which were not cytotoxic. The pro-apoptotic Bcl-2 family protein Bax translocates between the cytosol or mitochondria to regulate apoptotic sensitivity [51]. Further experimentation by Rudd and Devaraj showed that Bax translocation to mitochondria was markedly more efficient with C16:0 compared with C24:1 ceramide. Other CerS and their related ceramide molecular species have been linked with cancer, chemotherapy resistance, and metastasis [2,52,53]. In addition, very long-chain (C22 or greater) ceramides can exhibit actions distinct from corresponding saturated molecular species. For example, C24:1, has demonstrated a pro-survival role in ovarian cancer metastatic potential [54], whereas very long-chain dihydroceramides (C22:0; C24:0) were shown to be cytotoxic in T-cell acute lymphoblastic leukemia [55]. Demonstrating even more molecular species diversity, the increases in very long-chain ceramides (C24, C24:1, and C26:1) promoted oncogene-induced senescence in MCF10A mammary epithelial cells [56].

Note, however, that ceramides and S1-P are not the only sphingolipid metabolites to be considered in the context of dysfunctional sphingolipid metabolism and chemotherapy resistance; sphingomyelin, glycosylated sphingosine, sphingosine and sphinganine, and ceramide-1-phosphate may play yet to be revealed roles [34]. Glycosylated ceramide, as we shall discover, dominates a prominent position in chemotherapy resistance.

## 2. An Introduction to Chemotherapy Resistance

Drug resistance, a feature manufactured by the workings of biology and the ecosystem, is widespread and ever-present. Evolutionary forces such as selection pressure give rise to adaptations that enable protection from toxins, and chemotherapy drugs are no exception. Unfortunately, we are commonly confronted with drug-resistant bacteria and viruses. The H3N2 influenza A virus for example [57,58], pesticide-resistant insects that damage crops, and herbicide-resistant weeds, the blight of farmers; these all endanger human health. With specific regard to cancer, drug resistance is an all too frequent and unwelcome occurrence, as it is the major cause of cancer treatment failure.

Chemotherapy resistance can be intrinsic, as is characteristic in ovarian, pancreatic, and melanoma cancers. With intrinsic resistance, mechanisms preexist that render treatment ineffective. Drug resistance can also be acquired [59,60,61,62,63,64]; this type of resistance develops during treatment. In this scenario, patients initially respond and often experience periods of apparent, complete remission. One example of acquired resistance can be gleaned from the high-dose, poly-chemotherapy trials for the treatment of pediatric leukemia [65]. Early on, treatments were met with outstanding success; however, patients eventually relapsed with more aggressive, drug-resistant disease. Additional examples of intrinsic and acquired patterns of resistance include trastuzumab-resistant breast cancer [66] and therapies targeting BRAF and MEK to block the mitogen-activated protein kinase (MAPK) pathway, as seen in melanoma [67].

### 2.1. Factors Contributing to Chemotherapy Resistance

Many factors contribute to chemotherapy resistance (Figure 2). Some of the most notorious players include drug efflux driven by membrane transport proteins such as P-gp, drug activation/inactivation schemes, DNA damage repair, cell death inhibition, epigenetics [68], and resistance to targeted therapies [68]. This review will focus on the membrane protein P-gp, an ATP-binding cassette (ABC) transporter and its interplay with sphingolipid metabolism, which translates to diminished programmed cancer cell death.

### 2.2. Drug Activation/Inactivation—Drug Metabolism

Exerting a keen role in the development of chemotherapy resistance are metabolic processes that regulate drug activation. For example, cytarabine, a pyrimidine nucleoside analog, must be phosphorylated (activated) to be effective. Resistance to cytarabine occurs when tumor cells downregulate deoxycytidine kinase, the rate-limiting kinase involved in cytarabine activation, leading to the buildup of the inactive prodrug [69,70,71]. Juxtaposed to drug activation is drug inactivation. In this scenario, reduced glutathione (GSH) conjugation factors into the mix. GSH is the main intracellular low molecular weight thiol and nucleophilic scavenger. Maneuvers observed in resistance to platinum drugs comprise a crafty scheme. For example, GSH conjugation to platinum drugs generates products that are substrates for ABC transporters [72], discussed below. This is stealth drug resistance biology at its best, considering that platinum drugs are generally poor substrates for ABC transport. Also to be considered are isoforms of cytochrome and their overexpression and impact on drug metabolism and detoxification. Overexpression of cytochrome CYP1B1, for example, can enhance the biotransformation of numerous anticancer drugs, such as paclitaxel, mitoxantrone, and flutamide, and increased expression of another isozyme, CYP2A6, can catalyze the destruction of fluorouracil and cyclophosphamide [73].

### 2.3. Repairing DNA Damage

The mechanism of action of a myriad of chemotherapy drugs centers on DNA damage, either directly or indirectly, to target rapidly dividing cancer cells [74,75]. DNA damage repair, as in other mechanisms of drug resistance, protects cancer cells from what would otherwise be a debilitating endpoint. A good example of DNA damage repair occurs in the case of resistance to platinum drugs. Platinum-based anticancer agents are effective because they produce deleterious DNA cross-links, hampering cell division. Unfortunately, this “good” can be undone by nucleotide excision repair and homologous recombination, a DNA repair tool kit [76,77]. Many players are involved in the DNA damage response, as reviewed by Woods and Turchi [74].

### 2.4. Resistance to Apoptosis

Resistance to apoptosis constitutes another mechanism in the unwelcome repertoire of drug resistance. Anticancer agents, by and large, trigger apoptosis, a good thing to achieve therapeutic benefit. However, the inherent activation of antiapoptotic programs that give way to tumor cell proliferation, resistance, and disease recurrence is yet another undesirable cancer characteristic [78,79]. An orchestra of performers such as antiapoptotic Bcl-2 family proteins, autophagy, heat shock protein signaling, and the proteasome pathway contribute to apoptosis resistance in cancer, troublesome virtuosos playing busily like Mahler’s clamorous 8th symphony as opposed to a nocturne by Chopin. Dealing with such powerful biology may seem insurmountable; however, a broad range of strategies to overcome resistance to apoptosis are on the horizon [78,80,81]. These strategies present opportunities for exploration and exploitation. For in-depth reviews on chemotherapy resistance in cancer, the reader is directed to several excellent works [59,60,61,62,64,80,82,83].

### 2.5. Interplay between ABC Transporters and Sphingolipid Metabolism—Blunting of the “Ceramide Effect”

Upregulated expression of P-gp, one of several ABC transporter proteins, is perhaps the most consistent biological alteration in drug resistance biology [59,61,84,85,86]. Its overexpression is a hallmark of the multidrug-resistant phenotype. Members of this family include P-gp (*ABCB1*, previous gene symbol *MDR1*), MRP1 (multidrug resistance protein, *ABCC1*), and BCRP (breast cancer resistance protein, *ABCG2*). These proteins work to reduce the intracellular concentration of chemotherapy drugs via ATP-dependent drug extrusion, commonly termed “efflux”. This action limits therapeutic dosing and decreases the efficacy of substrate drugs. P-gp expression mediates intrinsic drug resistance in many chemotherapy-naive malignancies, such as breast, colorectal, renal, and ovarian cancers, neuroblastoma, acute myeloid leukemia (AML), non-Hodgkin’s lymphoma, and multiple myeloma [87]. Exposure to specific types of chemotherapy drugs, usually heterocyclics, upregulates P-gp expression, a scenario resulting in acquired drug resistance [88]. Beyond multidrug resistance, basal P-gp expression varied across various cancer types, with many groups reporting its upregulation and some reports of its downregulation [89]. In general, increased P-gp expression was associated with adverse outcomes. Despite the production of a wide array of ABC transporter antagonists designed to block chemotherapy efflux, clinical trials have been disappointing, providing little progress in combating resistance [90,91,92,93].

In Figure 2, “ABC Transporters” and “Dysfunctional Sphingolipid Metabolism” are joined, as we have recently shown drugs that upregulate P-gp expression, daunorubicin, for example, also upregulate sphingolipid enzyme expression and activity [35,36]. A myriad of ABC transporters, including P-gp, impact the metabolism of the sugar-containing glycosphingolipids such as GlcCer. These intrusions facilitate ceramide glycosylation, which, as we shall see, curtails apoptosis. Recent, outstanding works are available on the role of ATP-binding cassette transporters in the biosynthesis of glycosphingolipids [94,95]. To date, no P-gp inhibitor or sphingolipid-modulating drug has been approved specifically as an anticancer agent. For a detailed summary of the preclinical development and clinical evaluation of P-gp inhibitors [96] and sphingolipid-modulating drugs [97], we refer readers to the following comprehensive reviews.

## 3. Chemotherapy Drugs and Sphingolipid Metabolism

The impact of chemotherapy drugs on sphingolipid metabolism was first reported in landmark 1995 and 1996 papers by Bose et al. [98] in the USA and Jaffrezou et al. [99] in France. Both of these works showed that daunorubicin promoted ceramide formation in leukemia cells. In the work of Bose et al. [98], ceramide generation resulted from activation of de novo synthesis via CerS activity (see Figure 1). Fumonisin B1, a pan CerS inhibitor, diminished daunorubicin-governed ceramide generation and limited downstream apoptosis. Work by the Toulouse group in France [99] showed that daunorubicin-induced increases in ceramides, which preceded apoptosis, were not mediated by CerS but by neutral sphingomyelinase (Figure 1), a pathway that is unperturbed by fumonisin B1. These studies demonstrate the importance of ceramide generation in chemotherapy-induced apoptosis in cancer, in this instance by the anthracycline, daunorubicin, used in the treatment of leukemia. The ceramide flux origin differences between these two studies may have resulted from differences in daunorubicin concentrations used, cell types (e.g., human vs murine), and timing. With ceramide playing a major role in the induction of apoptosis in cancer cells [32,100], it is easy to envision that cancer cell defense mechanisms might employ stealthy metabolic maneuvers to regulate ceramide levels in a pro-survival direction; this idea is illustrated in the following section. One commonly utilized mechanism is upregulated ceramide glycosylation, which reduces the ceramide threat and is a hallmark of drug resistance in numerous cancer types [9,13,14,16,101,102,103].

### 3.1. Chemotherapy Selection Pressure Promotes Alterations in Sphingolipid Metabolism

Several studies have revealed that chemotherapy selection pressure (doxorubicin, daunorubicin, vinblastine, vincristine, cytarabine) alters the expression of key enzymes in sphingolipid metabolism [35,36,101,104]. With taxanes, modulation of sphingolipid metabolism accompanies paclitaxel resistance in a prostate cancer cell model that was marked by higher protein expression of SPHK1 and GCS and lower protein expression of acid sphingomyelinase and neutral sphingomyelinase 2, two enzymes involved in sphingomyelin catabolism and ceramide generation [105]. These data strongly suggest that chemotherapy resistance and sphingolipid metabolism have a cooperative relationship [102,106]. For example, work in AML demonstrated that HL-60 cells grown to accommodate daunorubicin coordinately upregulated P-gp expression and the expression of GCS, AC, SPHK1, and phospho-SPHK1 (Figure 3) [35]. Similar results were obtained with vincristine resistance [36]. Upregulation of these specific enzymes promotes antiapoptotic and mitogenic behavior and sabotages ceramide-mediated cell death. In this scenario, when sphingolipid-metabolizing enzymes are upregulated, the efficacy of ceramide-related therapies, ceramide nanoliposome (CNL) [107], for example, and ceramide-generating chemotherapy drugs (anthracyclines, fenretinide, etoposide) would be muted. Prominent mechanisms of drug resistance in AML are supported by diminished intracellular daunorubicin retention directed by P-gp, reduced capacity of the activating enzyme deoxycytidine kinase, and overexpression of the inactivating cytoplasmic 5′-nucleotidase II [108], the entry of sphingolipid metabolism into the blend adds a new dimension with possible therapeutic implications. On-point and as proof of principle, inhibition of sphingolipid-metabolizing enzymes has been shown to restore sensitivity to chemotherapy for a variety of cancers, as will be discussed below [109,110,111].

### 3.2. Ceramide Glycosylation—A Factor in Anthracycline and Vinca Alkaloid Resistance

Although AC, SPHK, and the ceramide-sphingosine-S1-P axis are important targets for regulating cancer cell growth [5,18,19,21,33,112,113,114,115,116,117,118], and their dysregulation in cancer makes for exciting research [119], herein we will focus on GCS, the enzyme catalyzing glycosylation of ceramide [120,121,122], and its role in chemotherapy resistance. Relatively little effort had been directed to the study of sphingolipids in drug resistance until the mid-1990s. Some of the first studies showed that multidrug-resistant cancer cells harbor high levels of GlcCer [9,123]. This trait has been observed in doxorubicin-resistant ovarian cancer cells, NCI-ADR/RES (Adriamycin-resistant cells formally known as MCF-7-AdrR cells), KB-V-1 adenocarcinoma (vinblastine-resistant HeLa subclone), NIH: OVCAR-3 cells that were established from a patient resistant to doxorubicin, in doxorubicin-resistant ovarian cancer cells, A2780AD, compared to the drug-sensitive, parental counterparts, along with AML cell lines described above [10,123,124]. Other studies showed that tumors from cancer patients who failed chemotherapy had high levels of GlcCer compared with specimens from chemotherapy-naïve patients [123].

Elevated GCS expression is associated with tumor progression and an overall reduction in intracellular ceramide levels, which thwarts the tumor-suppressor role of ceramide. Ceramide conversion to GlcCer via GCS confers “ceramide resistance” in cancer cells and contributes to the multidrug-resistant phenotype [8,16]. The activating influence of daunorubicin on CerS and sphingomyelinase elevates levels of proapoptotic ceramides [98,99], a good thing for anticancer treatment. These data support the idea that chemotherapy drugs that generate ceramide promote downstream upregulation of GCS. Interestingly, anthracyclines also upregulate the expression of P-gp; thus, GCS and P-gp are upregulated in tandem, as is often observed. Exposure to chemotherapy drugs in vivo produces similar upregulation of GCS. For example, ceramide levels in tumors from breast cancer patients were lower post-chemotherapy [125], implying that ceramide attrition is a response to drug treatment, and subsequent analysis revealed that ceramide levels were depressed via upregulated GCS expression. Further support is provided by a study showing that GCS is overexpressed in metastatic breast carcinoma [14]. Importantly, however, changes in ceramide mass may not be the quintessential element regarding cell fate, but instead, orchestrated, multi-phasic alterations in sphingolipid metabolism via GCS, AC, ceramide kinase, and sphingomyelin synthase may be the deciding factor.

### 3.3. Ceramide Glycosylation in Sorafenib, Imatinib, and Platinum Resistance

A role for ceramide metabolism in sorafenib activity is another example of the influence of ceramide glycosylation on drug resistance. Sorafenib blocks RAF kinase, a critical component of the RAF/MEK/ERK signaling pathway that controls cell division and proliferation. In studies in hepatocellular carcinoma cells, GCS was upregulated during sorafenib exposure and, more importantly, in sorafenib-resistant cell lines. Resistance was mitigated upon the inclusion of pharmacological inhibitors of GCS or through GCS silencing [111]. This therapeutic approach was also effective in vivo in a sorafenib-resistant tumor model, suggesting that targeting GCS may be a strategy for increasing sorafenib efficacy. Notably, GCS inhibition reversed sorafenib resistance and resulted in cytochrome c release and ATP depletion, a hallmark of ceramide’s elicitation of mitochondrial energetic collapse [126]. Targeting ceramide glycosylation has also been evaluated in imatinib resistance in a model of chronic myeloid leukemia. Imatinib inhibits the Bcr-Abl fusion protein tyrosine kinase, an abnormal enzyme produced by chronic myeloid leukemia cells that contain the Philadelphia chromosome, and also inhibits the receptor tyrosine kinases for platelet-derived growth factor. Results in this leukemia model demonstrated that chemical inhibition of GCS with D-*threo*-1-phenyl-2-decanoylamino-3-morpholino-1-propanol (PDMP) enabled ceramide accumulation and cell death in response to imatinib exposure [17].

In studies involving head and neck cancer, GCS inhibition improved cisplatin sensitivity [127]. These studies demonstrated that GCS and P-gp expression were associated with cisplatin resistance. Inhibition of GCS, using D,L-*threo*-1-phenyl-2-hexadecanoylamino-3-morpholino-1-propanol (PPMP), resulted in increased ceramide levels and augmented sensitivity to cisplatin in parental and in cisplatin-resistant in vitro and in vivo models. Although cisplatin has, in a few instances, been shown to increase intracellular ceramides [128,129], the findings of Roh et al. [127] are of interest because platinum drugs are poor substrates for the ABC transporters. Recent studies with oxaliplatin, the third-generation platinum drug, demonstrated that elevated GCS and GlcCer levels accompanied resistance in colorectal cancer cells, the first work to identify a role for ceramide glycosylation in oxaliplatin resistance [130]. This is yet another example of how elevated GCS activity enables tumor cells to maintain low levels of ceramide in response to the ceramide-generating capacity of oxaliplatin, thus abrogating the growth-inhibitory effects. Moreover, elevated *UGCG* gene expression correlated with decreased disease-free survival in colorectal cancer patients [130] and could provide a novel prognostic marker in colorectal cancer.

Although many studies point to the utility of GlcCer as an indicator of multidrug resistance, perhaps the bewildering question is why GlcCer accumulates in multidrug-resistant cancer and what are its ramifications? The role of P-gp in GlcCer accumulation in drug resistance will be discussed in the following section. Knowing what we know of ceramide’s tumor-suppressor properties, it is plausible that “hyper-glycosylation” arose as a check on ceramide-driven apoptosis (Figure 2). Because many anticancer agents are ceramide generators [12], ceramide glycosylation becomes a crafty maneuver to elevate chemotherapy resistance.

## 4. Manipulating GCS Modifies Response to Chemotherapy

Manipulating GCS activity up or down, pharmacologically or genetically, has been helpful in revealing the impact of ceramide glycosylation on response to certain anticancer drugs. In several in vitro cancer models, sensitivity to alkylating agents, anthracyclines, Vinca alkaloids, and taxanes could be increased or decreased by manipulating GCS expression and/or activity [16,103,109,131]. For example, transfection of drug-sensitive, wild-type breast cancer cells with GCS cDNA [132] provided resistance to doxorubicin [103]; a similar manipulation potentiated a resistance to TNF-α [133], which regulates apoptosis via ceramide generation from sphingomyelin [134]. These studies show that GCS overexpression alone is sufficient to confer drug resistance. Conversely, the downregulation of GCS expression in a chemotherapy-resistant cancer model increased cell sensitivity to anthracyclines, Vinca alkaloids, and taxanes [16]. Thus far, these studies place GCS in the role of sentry, overseeing the potential of ceramide to carry out an apoptotic program in response to anticancer drugs. In the same context, other studies have shown that limiting GCS activity decreases neuroepithelioma cell growth and reduces tumorigenicity in melanoma [135,136]. In melanoma, Deng et al. [135] successfully employed GCS antisense to inhibit tumor formation in an in vivo model, with a focus on ganglioside GM3 synthesis, a downstream product formed from lactosylceramide (LacCer), and work in multidrug-resistant MCF-7/AdrR (Adriamycin-resistant) cells (now termed NCI/ADR-RES cells) demonstrated elevated levels of GlcCer, LacCer, and gangliosides [9], compared with parental, drug-sensitive counterparts. Also, Hummel et al. [137] showed that transfection of HepG2 cells with *MDR1*/P-gp cDNA resulted in increased LacCer and ganglioside mass that was prompted by transcriptional upregulation of LacCer synthase, further works contributing to our knowledge of multidrug resistance-associated alterations in sphingolipid metabolism. By and large, these studies with multiple genetic manipulation models demonstrate the role of GCS in cancer cell response to heterocyclic chemotherapy drugs, the very drugs that promote the expression of P-gp and related drug pumps [138].

## 5. P-gp, GCS, and Ceramide Glycosylation—Cozy Cooperation

Curiously, GCS and P-gp overexpression have been shown to occur synchronously. In an Adriamycin-resistant breast cancer model, *MDR1*/P-gp knockdown increased cell death and was associated with decreased GCS mRNA expression [139]. Research in chemotherapy-resistant breast cancer, leukemia, melanoma, and colorectal cancer [101] also supports a view of a close protein partnership. Correspondingly, studies in an assemblage of multidrug-resistant adenocarcinoma cells, KB-V.01, KB-V.1, and KB-V1 [101], listed in order of increasing resistance to vinblastine, exemplify this partnership. In this model, GlcCer levels, GCS mRNA, GCS protein, and P-gp expression all increased with increased vinblastine resistance, suggestive of a GCS-P-gp “marriage” of sorts. It is within the realm of speculation to venture that because GCS is localized in the Golgi apparatus [140,141,142], wherein “non-plasma membrane-associated” P-gp has been shown to reside, these two proteins share in the control of ceramide glycosylation. Similar partnerships exist, as in sphingomyelinase synthase 1 complex formation with GCS, part of an important mechanism to regulate the metabolic fate of ceramide in the Golgi [143]. Therefore, chemotherapy selection pressure not only upregulates *MDR1*/P-gp [85,144,145] but also enhances the glycosylation of ceramide via GCS. Klappe et al. [146] and Kok et al. [147] reported similar findings in colchicine-resistant colon cancer. In that study, increases in GlcCer levels occurred concurrently with the upregulation of *MRP1*, a close cousin of *MDR1*/P-gp.

What might be the attraction in this cozy relationship between GCS and P-gp? Ceramide is a substrate for GCS, and the product of GlcCer can serve as a substrate for P-gp to mediate glycosphingolipid production [94]. In other words, “show me a substrate, and I will reveal a catalyst”. This substrate in search of a catalyst idea was demonstrated in work by Gouaze-Andersson et al. [15], who showed that exposure of cancer cells to GlcCer upregulated the expression of P-gp. This mechanism, known as enzyme auto-induction (by substrate), was shown in HepG2 cells exposed to high concentrations of resveratrol, wherein several resveratrol-detoxifying/metabolizing enzymes were induced [148].

How is GlcCer a substrate for P-gp? In work from the Netherlands, GlcCer was shown to flip into the Golgi by the catalytic “flippase” activity inherent in Golgi-resident P-gp [149]. Perhaps the presentation of an excess substrate (GlcCer) via the action of GCS signals for upregulated expression or stabilization of P-gp, as was shown by Gouaze-Andersson et al. [15], thereby enhancing the transport of GlcCer into the Golgi for anabolism to LacCer. Further insight into the function of sphingolipids, *ABCB1*/P-gp upregulation, and multidrug resistance is available in work by Lee and Kolesnick [150], who postulated that increased sphingomyelin content, developing early in some cancers, recruits and functionalizes plasma membrane *ABCB1*/P-gp, conferring a state of partial multidrug resistance that later brings glycosphingolipids into the picture.

Other work evaluating symbiosis between GCS and the ABC transporters comes from studies in a drug-resistant colon cancer cell model that showed GCS and P-gp levels decreased upon stable transfection of vincristine-resistant cells with GCS shRNA [151]. Importantly, this genetic manipulation reduced cell proliferation and restored vincristine sensitivity. Additional work showed evidence of a cooperative relationship whereby “ceramide resistance” is associated with increases in GCS, P-gp, and MRP-1 [11]. Work by Wegner et al. [152] complements the overall premise that GCS and drug resistance partner to propel the drug-resistant phenotype. In studies using GCS antisense and GCS inhibitors, both modalities dramatically depressed P-gp expression [153]. To summarize, the presumption that P-gp plays a role in ceramide glycosylation is substantiated by (*i*) work implicating GlcCer in the biochemistry of multidrug resistance [9], (*ii*) studies showing that elevated levels of GlcCer in cancer cells correlate with overexpression of P-gp [101], (*iii*) the finding that commonly employed P-gp antagonists suppress ceramide glycosylation in drug-resistant cancer cells [104], and (*iv*) insightful research showing that Golgi-resident P-gp can function as a sphingomyelin and GlcCer transmembrane flippase [149,154]. The pièce de resistance comes in elegant work from the Lingwood laboratory [95] that utilized photoreactive GlcCer analogs to identify GlcCer-binding protein specificity, work that pinpointed several of the ABC transporters, and strikingly, siRNA knockdown of the transporters differentially blocked glycosphingolipid synthesis.

Further work on the role of P-gp in the regulation of ceramide metabolism comes from Smyth et al. [155], who demonstrated that P-gp shielded chemotherapy-resistant leukemia cells from caspase-dependent apoptosis that was mediated by cytotoxic agents and by ligation of Fas, a ceramide-regulated death cascade. On a similar note, Shabbits and Mayer [156] showed that P-gp could regulate ceramide-mediated sensitivity to paclitaxel, a ceramide-generating taxane [157], implying that ceramide metabolism and apoptosis are regulated by both GCS and P-gp. The work of Morad et al. [158] strongly substantiates the cooperation of P-gp in the metabolism of ceramide by showing that multi-generational P-gp antagonists enhanced ceramide-based therapeutics via blockade of ceramide glycosylation with magnification of apoptotic responses. Lastly, the reader must consider the work of Sietsma et al. [159] on the role of sphingolipids in drug resistance. This insightful review, focusing on neuroblastoma, discusses the refined ensemble between sphingolipids and the ABC transporters that serve to support the drug-resistant phenotype.

In addition to ABC transporters, it is important to include the role of four-phosphate adaptor protein 2 (FAPP2) in glycosphingolipid metabolism, as this cytosolic transporter delivers GlcCer to the trans-Golgi network for globo-series glycosphingolipid biosynthesis [160,161]. Noteworthy, the knockdown of *ABCB1*/P-gp and *FAPP2* has distinct effects. Both can regulate glycosphingolipid metabolism in that both deletions were similarly effective in reducing cellular GlcCer and ganglioside levels; however, the specifics of these responses varied with cell type [95].

## 6. P-gp “The Chemotherapy Efflux Pump” versus P-gp “The Ceramide Neutralizer”—Versatility in Drug Resistance

Studies in wild-type and drug-resistant leukemia demonstrated that P-gp-expressing cells were resistant to ceramide-induced apoptosis; however, this resistance was averted by exposure to P-gp antagonists like Elacridar or cyclosporin A [162]. These results show that when P-gp activity was subdued, ceramide cytotoxicity ensued. This group also showed that when P-gp antagonists were present, rhodamine 123 (a P-gp substrate) accumulation in the Golgi was lost, and GlcCer formation was diminished. The idea that P-gp and other ABC transporters protect cancer cells from ceramide-orchestrated cytotoxicity is supported by studies in HeLa cells that conditionally express P-gp [163], a clean biological system for deciphering the role of P-gp in ceramide metabolism and consequent cytotoxicity. P-gp-expressing HeLa cells converted ceramide to GlcCer at four times the rate of the P-gp-deficient HeLa cells, a finding that implicates P-gp in the production of GlcCer. This action in P-gp-expressing cells serves to neutralize ceramide; strikingly, P-gp-expressing cells were resistant to ceramide, whereas P-gp-deficient cells were highly ceramide-sensitive. Interestingly, and as proof of principle, ceramide resistance in P-gp-expressing cells was reversed by exposure to P-gp antagonists, and this was accompanied by a block in GlcCer production. These works demonstrate that P-gp can function as a “ceramide neutralizer”, suggesting new and novel indications for P-gp antagonists in ceramide-centric therapeutics [158]. The role of P-gp in ceramide neutralization was further explored in work by Chapman et al. [164], who demonstrated that multidrug-resistant, P-gp-expressing ovarian cancer cells have an enhanced capacity for conversion of ceramide to GlcCer while the levels of GCS enzyme activity, measured in cell-free assays, were the same. Thus, when you take the cell apart, the differences in ceramide glycosylation rates even out, and the need for the intact “P-gp/Golgi machinery” of the cell becomes apparent.

## 7. P-gp Antagonists and GCS Inhibitors—Duplicitous Roles

We propose that P-gp antagonists serve dual roles in cancer therapy. Firstly, they block the efflux of anticancer agents by the ABC transporters, and secondly, they halt the conversion of ceramide to GlcCer, thereby fortifying the intrinsic pathway of ceramide-governed, mitochondrial-driven apoptosis [32]. This scenario is magnified when ceramide-generating drugs are used (anthracyclines, for example [12]) and with ceramide delivery platforms (ceramide nanoliposome (CNL), for example [107]). This scheme is illustrated in Figure 4. Daunorubicin and other drugs (e.g., imatinib, fenretinide, cytarabine, etoposide, CNL) can activate de novo and SMase-driven pathways of ceramide generation to induce mitochondrial-associated apoptosis. GCS inhibitors and P-gp antagonists block ceramide glycosylation directly at GCS or indirectly at Golgi-resident P-gp, respectively, by disrupting GlcCer entry into the Golgi and eliciting product inhibition of GCS. Importantly, Golgi membranes are high in P-gp [165]. Taken together, the anticancer properties and the ceramide-generating capacity of daunorubicin and other anticancer therapeutics (e.g., doxorubicin, paclitaxel, etoposide, imatinib, fenretinide, resveratrol, cytarabine, ionizing radiation, and CNL) can be utilized, and enhanced by a wide range of P-gp antagonists such as ketoconazole, tamoxifen, verapamil, cyclosporin A, VX-710, PSC833 (Valspodar), zosuquidar, and certain GCS inhibitors like PDMP. Indeed, we recently demonstrated that both P-gp antagonists (verapamil and zosiquidar) and the GCS inhibitor PDMP enhanced cell death mediated by AC inhibition in drug-resistant AML [166]. Of note, however, pharmacokinetic considerations may factor in and this is to be considered and researched. P-gp antagonists have a poor history in the clinic together with cytotoxic chemotherapy, but combinatorial use with sphingolipid targeting therapies has yet to be explored [91].

What role might GCS inhibitors play in chemosensitization? Studies in P-gp-expressing leukemia cells demonstrated that the GCS inhibitor PDMP elicited sensitization to daunorubicin via (*i*) efflux blockade, (*ii*) ceramide accumulation, and (*iii*) reduction in GlcCer levels [167], a clear example of the duplicitous role exerted by agents like PDMP. In other studies, PDMP and tetrandrine, a P-gp antagonist, exerted similar effects in multidrug-resistant leukemia cells [168].

## 8. Do We Target GCS or P-gp to Enhance the Ceramide Effect?

Experiments aimed at defining potential, off-target intracellular “receptors” for the GCS inhibitors yielded interesting findings. The high-affinity GCS inhibitor, ethylenedioxy-1-phenyl-2-hexadeca-noylamino-3-pyrrolidino-1-propanol (ethylenedioxy-P4) [169], at a low dose (0.2 µM), blocked the conversion of C6-ceramide to C6-GlcCer by >90% in doxorubicin-resistant, P-gp-rich ovarian cancer cells, but that concentration did not enhance C6-ceramide cytotoxicity [164], a suggestion that inhibition of GCS is not what matters. Curiously, when the concentration of ethylenedioxy-P4 was increased to 5.0 μM, C6-ceramide cytotoxicity proceeded. These data suggest that at higher doses, GCS inhibitors interact with P-gp similarly to true P-gp antagonists. Therefore, it may be more beneficial to control ceramide flipping at P-gp rather than GCS-mediated ceramide glycosylation. The polygamous nature of GCS inhibitors was further revealed in the work of Sietsma et al. [170], who demonstrated that PDMP decreased paclitaxel and vincristine efflux in neuroblastoma cells, thus acting as a P-gp antagonist. The pièce de resistance came from similar studies by Chai et al. [171], who showed that the chemotherapy-sensitizing properties of Genz-123346, a potent GCS inhibitor, were mediated mainly through modulation of P-gp function. These authors showed that GCS inhibitors were substrates for the drug resistance efflux pumps. Perhaps, then, P-gp helps detoxify ceramide by promoting glycosylation, but when P-gp is blocked, ceramide transfer and flipping at the Golgi halts, and ceramide potency is augmented, features previously ascribed to GCS inhibition. This concept was bolstered by work showing that inhibition of GCS was not sufficient to reverse drug resistance in cancer cells [172]. So, why does that study’s conclusion stray from the concepts related herein? Importantly, these investigators employed the GCS inhibitor, NB-DNJ (N-butyl-deoxynojirimycin, miglustat), a water-soluble imido sugar with no affinity for P-gp; chemically speaking, NB-DNJ is neither a substrate for P-gp nor a P-gp antagonist. This again suggests that inhibition of GCS is not what matters, but rather inhibition of flux through the ceramide glycosylation pathway matters, by whatever means we can achieve it. Others have also demonstrated the differential impact of glycolipid biosynthesis inhibitors, PDMP and NB-DNJ, on ceramide generation and cell death in cancer models [173].

The indications for P-gp antagonists and GCS inhibitors in cancer therapy are summarized in Table 1. These data set forth a unique juxtaposition, demonstrating cooperation between P-gp antagonists and GCS inhibitors, wherein their employ overlaps and is beneficial in enhancing the action of a myriad of anticancer drugs. For example, increased sensitivity to proapoptotic C6-ceramide can be accomplished in leukemia and colorectal cancer cells via the inclusion of either tamoxifen, tariquidar, or zosuquidar, first- and third-generation P-gp antagonists (Table 1C) [158]. PDMP, a GCS inhibitor, and tetrandrine, a P-gp antagonist, were both effective in reversing daunorubicin resistance in a model of drug-resistant leukemia (Table 1G), and both agents enhanced daunorubicin accumulation, evidence that GCS inhibitors have an affinity for P-gp [171]. In neuroblastoma (Table 1I), another example of duplicitous roles is exemplified by the ability of GCS and P-gp antagonists to increase Taxol and vincristine sensitivity while decreasing Taxol and vincristine efflux [170].

Table 1 also lists many examples of the beneficial impact of PDMP on reversing resistance to a wide range of anticancer agents (e.g., imatinib, cisplatin, fludarabine). Moreover, the Bcl-2-family inhibitors, ABT-737 and ABT-263, and PDMP (Table 1Q,R) also demonstrate overlapping actions. For example, in drug-resistant lung adenocarcinoma, PDMP and ABT-737 enhance sensitivity to vinorelbine, a ceramide-generating drug [179], while PDMP sensitizes human leukemia cell lines to ABT-263 [180]. As ABT-737 and ABT-263 are substrates for P-gp [188], in addition to inhibition of Bcl-2 family proteins, we can envision enhanced ceramide levels as well as diminished drug efflux coming into play in vinorelbine sensitization.

An interesting study with GNF-2 (Table 1U), a tyrosine kinase inhibitor developed to overcome imatinib-resistant mutations found in chronic myeloid leukemia (CML) patients, showed that PDMP sensitized a highly drug-resistant CML model, CML-T315I, to this Bcr-Abl inhibitor. PDMP also increased sensitivity to imatinib and nilotinib [183]. As imatinib resistance is associated with overexpression of P-gp in CML patients [189], it is plausible that PDMP efficacy in this instance is supported by blocking imatinib efflux. On point, these tyrosine kinase inhibitors interact with P-gp [190], and imatinib has been shown to induce apoptosis in leukemia cells via the generation of C18-ceramide [191]. Regarding the further mechanism of action for this combination, cytotoxicity of the tyrosine kinase inhibitors is perhaps intensified by imatinib-generated ceramide, blockade of ceramide glycosylation by PDMP and/or imatinib, and possible diminution of drug efflux by PDMP; a complicated orchestration, to say the least.

Mifepristone, the antiprogestin also known as RU-486, can enhance vincristine and Adriamycin sensitivity in drug-resistant gastric cancer, hepatoma, and leukemia cells (Table 1V) [184,185], and fittingly, mifepristone is a P-gp antagonist [185] as well as a ceramide glycosylation blocker in drug-resistant cells [192]. Lastly, resveratrol, the polyphenolic found in red grapes, potentiates Taxol and doxorubicin cytotoxicity (Table 1W) via its ability to inhibit P-gp-governed drug efflux and through downregulation of *MDR1*/P-gp expression [186]. Here, the double entendre is that resveratrol is also a ceramide generator, as shown in CML cells [193] and in triple-negative human breast cancer cells [194]. So, an agent like resveratrol exhibits versatility in the form of P-gp antagonism as well as ceramide generation, and if used in conjunction with daunorubicin, benefits should be synergistic in drug-resistant cancer.

## 9. Summary and Conclusions

Although it may be an over-simplification, the take-home message from many studies is that P-gp (and perhaps other ABC transporters) confers resistance to ceramide-directed apoptosis by enhancing ceramide glycosylation. In our conversations, Cliff Lingwood posed that “several, if not all, ABC transporters, and particularly those that move lipid-soluble substrates, can flip GlcCer to generate different pools for differential glycosphingolipid biosynthesis”. Thus, ABC transporters provide a new target and a smorgasbord of possibilities for the regulation of glycosphingolipid biosynthesis. Perhaps the efficiency of this flipping contributes to ceramide resistance by providing a driving force, like a conveyer belt, to push ceramide towards glycosylation. Drug resistance, in this instance, can be to ceramide if used directly as a therapeutic [195] or to the many chemotherapy drugs that generate and utilize ceramide in signaling apoptosis [12]. Thus, selection pressure for resistance to anticancer drugs not only upregulates *MDR1*/P-gp-mediated drug efflux [85,144,145] but also its enhancement of ceramide metabolism via glycosylation. These actions, in conjunction with GCS upregulation, should certainly lead to ceramide attrition, a condition that would avert tumor cell apoptosis.

In closing, this review presents a somewhat perplexing story about cancer’s “bad guy”, P-gp, and new mechanisms whereby P-gp inhibitors can be utilized in a therapeutically beneficial manner (Figure 4). For example, ample data show that ceramide-governed tumor cell death pathways can be intensified by blocking ceramide glycosylation through the use of P-gp antagonists [158,164,196]. In addition, we see that the use of GCS inhibitors like PDMP, which subvert Taxol and vincristine efflux [170], can be beneficial. Also, in the plus category, GCS inhibitors sensitize cancer cells to chemotherapy by sustaining ceramide levels promoted by ceramide-generating drugs, and they also act to suppress P-gp expression. Although there are no reported clinical studies evaluating P-gp inhibitors or GCS inhibitors in the context of improving ceramide therapies, this array of activities presents us with interesting possibilities for exploring chemotherapy resistance from a different angle.

## Figures and Tables

**Figure 1 ijms-25-09825-f001:**
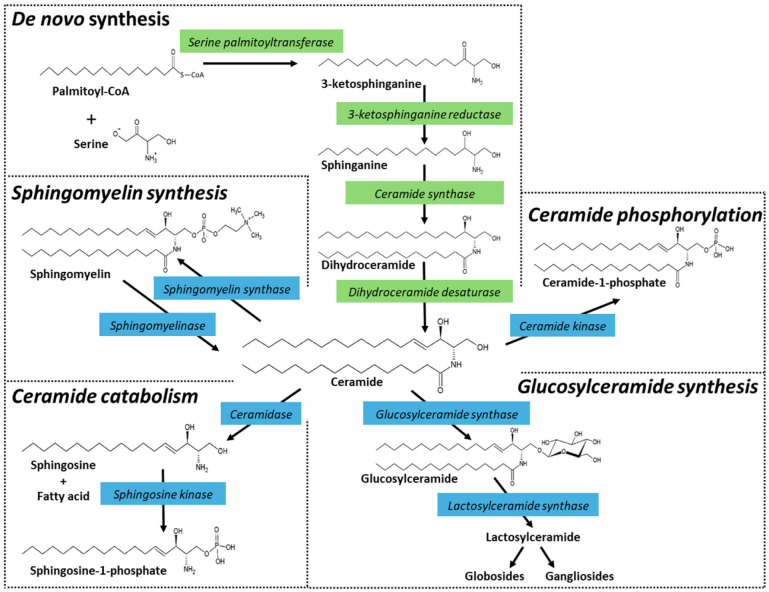
Ceramide metabolism and metabolism of higher sphingolipids. Ceramide is synthesized in the endoplasmic reticulum. The de novo biosynthesis starts with the condensation of serine and palmitoyl-CoA, catalyzed by serine palmitoyltransferase. The product, 3-ketosphinganine, contains 18 carbons and is reduced to sphinganine by 3-ketosphinganine reductase. The next step generates the saturated ceramide precursor, dihydroceramide, via the action of ceramide synthases, of which there are six isoforms (CerS1-6) that ultimately give rise to a multitude of molecular species with distinct roles. Finally, although dihydroceramide is nearly identical in structure to ceramide, it lacks the 4,5-*trans* double bond, which is inserted by dihydroceramide desaturase to form ceramide. Ceramide can also be produced by the action of specialized phospholipases, known as sphingomyelinases. Sphingomyelinases, which are characterized according to their pH optimum and subcellular locations, cleave sphingomyelin at the phosphodiester bond that is proximal to ceramide, producing ceramide and choline phosphate. Ceramide can also be formed by the action of ceramide-1-phosphate (C1-P) phosphatase and by glucosylcerebrosidase. Once produced, ceramide can be hydrolyzed by ceramidases, glycosylated by glucosylceramide synthase (GCS), used to generate sphingomyelin by sphingomyelin synthases, or phosphorylated by ceramide kinase producing C1-P. Strategic points in de novo synthesis and in subsequent ceramide metabolism can be activated or inhibited, providing useful avenues to define ceramide-regulated events, such as cell fate. Enzyme inhibitors and P-glycoprotein (P-gp) antagonists are often used to amplify the induction of cell death by ceramide. Enzymes that deplete intracellular ceramide either by catabolism (e.g., ceramidase activity) or anabolism (e.g., glycosylation, sphingomyelin synthesis, phosphorylation) can contribute to cancer cell growth.

**Figure 2 ijms-25-09825-f002:**
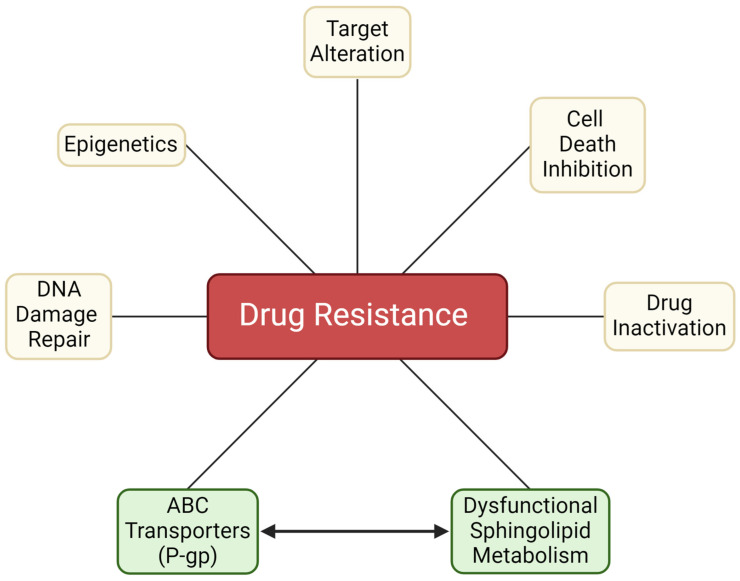
Factors contributing to chemotherapy resistance. The inclusion of dysfunctional sphingolipid metabolism marks an important biology that contributes to chemotherapy resistance. As noted in the text, modifications in sphingolipid metabolism work in concert with other contributors, such as the ABC transporter, P-gp, which directs chemotherapy efflux and also participates in ceramide metabolism. P-gp, P-glycoprotein. Created with BioRender.com.

**Figure 3 ijms-25-09825-f003:**
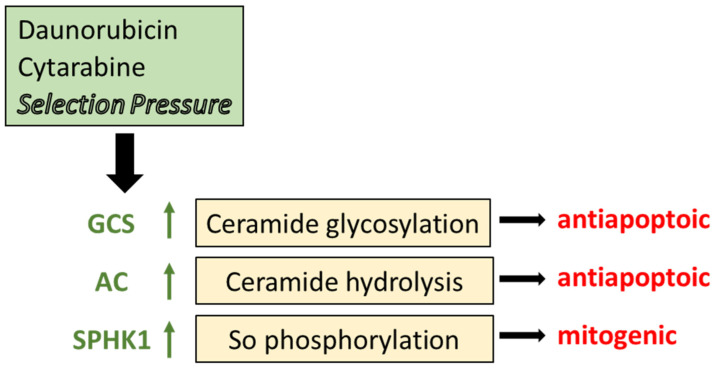
The impact of chemotherapy selection pressure on sphingolipid metabolism. Cancer cells grown to acquire resistance to anticancer agents display prominent elevations in the expression and activity of some of the key enzymes in sphingolipid metabolism, including GCS, AC, and SPHK1 as indicated by the upward arrows. Through glycosylation, GCS converts proapoptotic ceramide to GlcCer, a detoxified product. Likewise, AC hydrolyzes ceramide to disable its cytotoxic impact. SPHK1 utilizes sphingosine, a product of AC, to produce S1-P and elicit mitogenic responses. GCS, glucosylceramide synthase; AC, acid ceramidase; SPHK1, sphingosine kinase 1; So, sphingosine.

**Figure 4 ijms-25-09825-f004:**
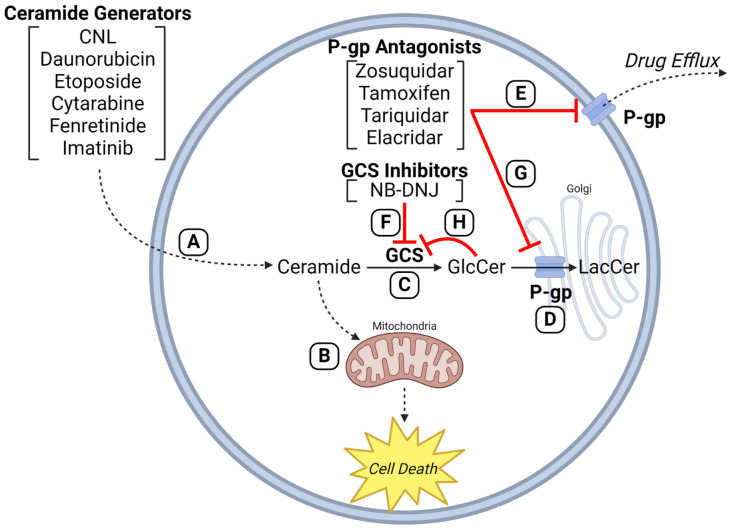
P-glycoprotein antagonists enhance efficacy of anticancer agents that employ ceramides in mechanism of action. (**A**) Anticancer drugs (e.g., CNL, daunorubicin, etoposide, cytarabine, fenretinide, imatinib) can increase intracellular ceramide levels via de novo synthesis, ceramide synthase activation, and sphingomyelinase activation. (**B**) The resulting ceramide deluge converges on mitochondria to elicit the intrinsic pathway of apoptosis (pro-death). Upregulated activities of (**C**) GCS and (**D**) P-gp, characteristic in multidrug resistance, hijack this ceramide-governed cell death pathway by hyper-conversion of ceramide to GlcCer. (**E**) P-gp antagonists block drug efflux to circumvent efflux-mediated resistance mechanisms. (**F**) GCS inhibitors (e.g., NB-DNJ) and (**G**) P-gp inhibitors (e.g., zosuquidar, tamoxifen, tariquidar, elacridar) can block ceramide glycosylation directly at GCS or indirectly at Golgi-resident P-gp by blocking GlcCer entry into the Golgi, promoting buildup of GlcCer and product inhibition of GCS. (**H**) The inhibitory symbol from GlcCer to GCS indicates product inhibition. GCS, glucosylceramide synthase; P-gp, P-glycoprotein; GlcCer, glucosylceramide; NB-DNJ, 1N-Butyldeoxynojirimycin; LacCer, lactosylceramide; CNL, ceramide nanoliposome. Created with BioRender.com.

**Table 1 ijms-25-09825-t001:** Therapeutic strategies to combine P-pg antagonists and GCS inhibitors in cancer.

Action	Agents	Model	Reference
**A**. Increase Adriamycin sensitivity	Tamoxifen; PPMP	MCF-7/AdrR cells (Adriamycin-resistant, human ovarian cancer cell line)	[104]
**B**. Increase Taxol, vincristine sensitivity	PDMP	MCF-7/AdrR	[156]
**C**. Increase C6-ceramide sensitivity	Tamoxifen; tariquidar; zosuquidar; cyclosporin A	HL-60/VCR (vincristine-resistant human acute myeloid leukemia cell line); KG-1a (human acute myeloid cell line); LoVo (human colon cancer cell line)	[158]
**D**. Enhance C8-ceramide sensitivity	Elacridar; cyclosporin A	KG-1a; TF-1 (human acute myeloid leukemia cell lines)	[162]
**E**. Increase C6-ceramide sensitivity	Tamoxifen	P-gp-overexpressing HeLa cells	[163]
**F**. Increase daunorubicin and vincristine sensitivity; increase rhodamine retention	PDMP	KG-1a/200 (daunorubicin- and vincristine-resistant); K562/138 (vincristine-resistant chronic myeloid leukemia cell line); K562/MDR-1 (stably transfected to express MDR-1)	[167]
**G**. Reverse daunorubicin resistance; enhance daunorubicin accumulation	PDMP; tetrandrine	K562/A02 (Adriamycin-resistant)	[168]
**H**. Enhanced C6-, C8-ceramide cytotoxicity	Tamoxifen; cyclosporin A; VX-710 (Biricodar); P4 (analog of PPPP)	2780AD (doxorubicin-resistant human ovarian cancer cell line); NCI/ADR-RES (doxorubicin-resistant, human ovarian cancer cell line)	[164]
**I**. Increase sensitivity to Taxol and vincristine; decrease efflux	PDMP; PSC833 (Valspodar); MK571	Neuro-2a; C1300 (murine neuroblastoma cell lines)	[170]
**J**. Sensitize drug-resistant cells to chemotherapy (Vincas; Adriamycin; Taxotere)	Genz-123346; PDMP; verapamil	KBV-1 (human cervical carcinoma cell line); HCT-15 (human colorectal cancer cell line)	[171]
**K**. Reverse Imatinib resistance	PDMP	K562/IMA (imatinib-resistant chronic myeloid leukemia cell line)	[17]
**L**. Reverse Cisplatin resistance	PDMP	Orthoxenografts, testicular germ cell tumors	[174]
**M**. Reverse Fludarabine resistance	PDMP	Fludarabine-resistant MEC-2 cells (human B cell leukemia cell line)	[175]
**N**. Enhance sensitivity to Taxol and cisplatin	PDMP	Paclitaxel-resistant PX2 and KF28; cisplatin-resistant C13 and KFr13 (human ovarian cancer cell lines)	[176]
**O**. Enhance selumetinib (AZD-6244) sensitivity	PDMP	PANC-1; AsPC-1; MIA PaCa-2 (human pancreatic cancer cell lines)	[177]
**P**. Increase sensitivity to C6-ceramide nanoliposome (CNL)	PDMP	PANC-1	[178]
**Q**. Enhance vinorelbine sensitivity	PDMP	A549; CL1-5 (human lung adenocarcinoma cell lines)	[179]
**R**. Sensitization to ABT-263 (Bcl-2 inhibitor)	PDMP	U937; K562; HL-60; RPMI-8226 (various human leukemia cell lines)	[180]
**S**. Enhance AR-42 (HDAC inhibitor) sensitivity	PDMP	SW-620 (human colon cancer cell line)	[181]
**T**. Potentiate A-674563 (Akt1 inhibitor) cytotoxicity	PDMP	A375 (human melanoma cell line)	[182]
**U**. Sensitize to GNF-2 (Bcr-Abl inhibitor); imatinib; nilotinib	PDMP	K562; Ba/F3-p210^T315I^ (Bcr-Abl mutant murine chronic myeloid leukemia cell line)	[183]
**V**. Enhance vincristine and Adriamycin sensitivity; decrease drug efflux	Mifepristone	SGC7901/VCR (vincristine-resistant human gastric cancer cell line); rat hepatoma cells; K562-R7 (multidrug-resistant)	[184,185]
**W**. Enhance sensitivity to Taxol and Adriamycin	Resveratrol	MCF-7; HepG2, HeLa (various human solid tumor cell lines)	[186]
**X**. Enhance vincristine, daunorubicin, cisplatin sensitivity	Ethylenedioxy-P4	Lucena-1 (vincristine-resistant K562); FEPS (daunorubicin-resistant K562)	[131]
**Y**. Enhance SACLAC (acid ceramidase inhibitor) sensitivity	PDMP	HL-60/VCR (vincristine-resistant); HL-60/DNR (daunorubicin-resistant)	[166]
**Z**. Enhance hydroxychloroquine or DC661 sensitivity (lysosomal autophagy inhibitors)	Genz-123346	A375-P (human melanoma cell line); DLD-1 (human colorectal adenocarcinoma cell line); B16-F10 (murine melanoma cell line); MIA PaCa-2; A549; PDXWM4380 and PDX WM4552 (patient-derived cell lines)	[187]
**AA**. Reverse sorafenib resistance	PDMP	Hep3B; HepG2 (hepatocellular carcinoma cell lines)	[111]
**AB**. Sensitize to cisplatin	PPMP	HN9-cisR (cisplatin-resistant human head and neck cancer cell line)	[127]

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
