# Peer review of "The Drug Transporter P-Glycoprotein and Its Impact on Ceramide Metabolism—An Unconventional Ally in Cancer Treatment"

_ijms, 2024, doi:10.3390/ijms25189825_

Round 1
Reviewer 1 Report
Comments and Suggestions for Authors
This manuscript provides a comprehensive analysis of the involvement of P-glycoprotein (P-gp) in ceramide metabolism and its significance in cancer therapy. The authors prepared a comprehensive study of the impact of P-gp, a protein commonly associated with drug resistance, on sphingolipid metabolism, particularly ceramide glycosylation, and its effect on cancer cell fade. The review is meticulously organized, encompassing the intricate processes of sphingolipid metabolism, the underlying reasons of chemotherapy resistance, and the promising possibilities of therapeutic approaches.
The following points may enhance the manuscript:
Readers unfamiliar with lipid metabolism or cancer biology may find the extensive use of technical terminology in the book to be a barrier to understanding. Please provide clearer explanations of complex sentences and language and streamlining key portions could enhance accessibility.
There are bolded and underlined words, eg symphony, daunarubicin, ceramide, why???
The font not similar across the manuscript, check it.
More detailed annotations and shorter explanations would help readers understand the complex metabolic pathways described in the manuscript's figures.
The manuscript cites a considerable amount of research, however it would be advantageous to incorporate more recent findings, specifically those published within the past two years, in order to ensure the review is current.
- A more detailed comparative analysis of different P-gp inhibitors and their efficacy in various cancer models could provide valuable insights for researchers and clinicians.
- More thorough information about ongoing or completed clinical trials involving P-gp antagonists and ceramide-based medications will improve the therapeutic applications discussion. This might help clarify whether or not the suggested methods have any real-world applications.
- Providing a comprehensive analysis of the potential adverse effects and difficulties related to the use of P-gp antagonists and GCS inhibitors in clinical settings will present a more equitable viewpoint on the suggested treatment strategies.
- The manuscript contains minor typographical errors that need correction to enhance its professional presentation.
- Ensuring that all references are current and accurately cited is crucial. Cross-referencing with recent publications could enhance the credibility of the review.
- Simplify technical language where possible and provide clearer explanations of complex terms and pathways.
- Improve the clarity of figures and diagrams with detailed annotations and simplified descriptions.
- Incorporate recent studies and findings to ensure the review is current and comprehensive.
- Include a more detailed comparative analysis of different therapeutic approaches involving P-gp and ceramide metabolism.
- Discuss potential side effects and challenges associated with the therapeutic use of P-gp antagonists and GCS inhibitors.
- Some sentences are overly complex, making them difficult to follow. Breaking down long sentences into shorter, more manageable ones can improve readability.
Example: "These data suggest that at higher doses, GCS inhibitors interact with P-gp in much the same fashion as true P-gp antagonists, thus it seems there is more benefit to be derived from controlling ceramide flipping at P-gp as opposed to GCS-mediated ceramide glycosylation."
Improved: "These data suggest that at higher doses, GCS inhibitors interact with P-gp similarly to true P-gp antagonists. Therefore, it may be more beneficial to control ceramide flipping at P-gp rather than GCS-mediated ceramide glycosylation."
- The manuscript contains redundant phrases and wordy expressions that could be streamlined for clarity.
- Example: "On point, regarding the role of P-gp in ceramide metabolism and ceramide cytotoxicity, are studies in wild-type and in drug-resistant leukemia that demonstrated P-gp-expressing cells were resistant to ceramide-induced apoptosis; importantly, however, this resistance was averted by exposure to either Elacridar or cyclosporin A, P-gp antagonists."
- Improved: "Studies in wild-type and drug-resistant leukemia demonstrated that P-gp-expressing cells were resistant to ceramide-induced apoptosis; however, this resistance was averted by exposure to P-gp antagonists like Elacridar or cyclosporin A."
- While the use of technical terminology is necessary, ensuring that these terms are clearly defined and explained, especially for a broader scientific audience, can enhance comprehension.
- Example: "The balance between ceramide and S1-P is critical in the control of cancer cell fate."
- Improved: "The balance between ceramide and sphingosine-1-phosphate (S1-P) is critical in controlling cancer cell fate."
- Conduct thorough proofreading to correct typographical errors and grammatical issues.
Comments on the Quality of English Language
The manuscript demonstrates a good command of English, with a professional and academic tone suitable for a scientific journal. However, there are several areas where the quality of the language can be improved to enhance clarity, readability, and overall presentation. Below are detailed observations and recommendations for improving the quality of the English language used in the manuscript.
Reviewer 2 Report
Comments and Suggestions for Authors
The manuscript provides a comprehensive overview of the role of ceramide in chemotherapy and its interaction with P-glycoprotein (P-gp) in cancer drug resistance. It presents a well-rounded discussion of the topic, although there are areas that could benefit from increased clarity, conciseness, and grammatical precision.
The manuscript contains long, complex sentences that could be simplified to improve readability. Breaking down these sentences would help convey the information more clearly.
Example: "These drugs can activate de novo synthesis of ceramide or stimulate production of ceramide via sphingomyelinases, all good events to limit cancer cell survival." This could be rephrased for simplicity.
The use of informal terms like "highjacks" should be replaced with more formal language to maintain academic tone.
Example: Replace "highjacks" with "disrupts" or "interferes with".
The manuscript covers a significant and well-researched topic but can benefit from enhanced clarity, structure, and conciseness. By addressing these areas, the manuscript will be more accessible and impactful to its intended audience.
Reviewer 3 Report
Comments and Suggestions for Authors
Reviewer report
The drug transporter P-glycoprotein and its impact on ceramide metabolism—an unconventional ally in cancer treatment
This is a well-organized study showing the importance of pgp and sphingolipid metabolism and its dysfunctional regulation in cancer. The article focuses lights on sphingolipid metabolism and impact of chemotherapy selection on sphingolipid metabolism as well as link between pgp. Articles needs to be improved in some aspects.
I. Please describe the entire article in the form of a graphical abstract.
II. This manuscript doesn't follow font uniformity as per the journal recommendation please make necessary corrections.
III. A few sentences have been made to bold font please correct the sentences.
IV. How does P-gp is related to tumor immunity please explain.
V. How does pgp expression vary as per the tumor type please explain.
VI. Can authors list out clinical studies related to the current topic in the form of table?
VII. Please describe pre-clinical studies related to pgp in the form of a table.
Round 2
Reviewer 1 Report
Comments and Suggestions for Authors
The revised manuscript provides an in-depth review of the role of P-glycoprotein (P-gp) in ceramide metabolism and its implications for cancer treatment. The article effectively discusses the dual role of P-gp in mediating drug resistance and its interaction with ceramide metabolism, emphasizing the potential therapeutic avenues for targeting these pathways in cancer treatment.
While the revised manuscript is much improved, there are still a few areas where further enhancement could be beneficial:
- Consistency in Terminology: Ensure consistent use of terminology throughout the manuscript. For instance, the terms "ceramide glycosylation" and "GlcCer production" are used interchangeably. It would be clearer if one term is consistently used or if a brief explanation is provided to clarify that they refer to the same process./span/p
p class="MsoListParagraphCxSpMiddle" style="text-align: justify; text-indent: 0pt"- Abstract Revision: The abstract could be slightly condensed to better highlight the key findings and implications of the review. Consider focusing on the novel insights provided by the review and their potential impact on cancer therapy./span/p
p class="MsoListParagraphCxSpMiddle" style="text-align: justify; text-indent: 0pt"- Conclusion Section
Comments on the Quality of English Language
Minor editing of English language is required. Author Response Please see the attachment.
